# The Royal Chapel of Pedro I of Castile in the Christianised Mosque of Seville

Pablo Gumiel-Campos [1,2]

1    Institute of Art History, Universidade Nova de Lisboa, 1099-085 Lisbon, Portugal; gumiel.pablo@gmail.com
2    Department of Art History and Theory, Universidad Autónoma de Madrid, 28049 Madrid, Spain

**Abstract:** Pedro I of Castile (1350–1369) founded a royal chapel in the Christianised Mosque of Seville. He intended to house there his body, that of Queen María de Padilla, and their son the Infant Alfonso (1359–1362). This mausoleum is well documented both in the king's will and in the chronicles of López de Ayala; however, there are no material remains as it was demolished with the construction of the new cathedral in the 15th century. In this article, we seek to produce a state of the art history of the building, a compilation of all the documentary sources that exist for its analysis, and an approach to the problems that hinder its study. We have also tried, unsuccessfully, to put forward a hypothesis about its original location, but we have come up against a dead end. Despite this, we consider it essential to lay all the cards on the table and prevent the mausoleum from falling into oblivion.

**Keywords:** Pedro I; Royal Chapel; Seville; Castille; Christianised Mosque of Seville; Maqṣūra; lost monuments; architecture; late middle ages

## 1. Introduction and State of the Art about the Royal Chapel of Pedro I

Pedro I of Castile founded, by testamentary order, a royal chapel in the Christianised Mosque of Seville. He intended to house there his body, that of Queen María de Padilla, and their son the Infant Alfonso. This mausoleum is well documented; however, there are no material remains as it was demolished with the construction of the new Gothic cathedral in the 15th century. The purpose of the article is to produce a state of the art history of the building, a compilation of all the documentary sources that exist for its analysis, and an approach to the problems that hinder its study. We have also tried to put forward a hypothesis about its original location, but we have come up against a dead end. Nevertheless, it is essential to lay all the cards on the table and prevent the mausoleum from falling into oblivion.

One of the first historians to note the existence of the Royal Chapel founded by Pedro I in the original cathedral of Seville was Juan Antonio de Vera y Figueroa (1647), who stated: "[Pedro I] founded a distinguished chapel in Seville, which he enriched with ornaments (he made liberal donations and endowed with twelve chaplaincies that always supported his soul)"[1]. Nearly two centuries later, in 1831, Picado Franco (1831, p. 106) transcribed the king's will recovering the news, without paying too much attention to the information regarding the foundation of the chapel. It was not until the very end of the 20th century that the building was mentioned again.

In 1998, Professor Laguna Paúl (1998, p. 54) noted the existence of documentary references to a royal chapel commissioned by Pedro I, but which had not yet been located. Laguna Paúl's publication, entitled *La Aljama cristianizada. Memoria de la catedral de Santa María de Sevilla* was part of a series of studies that aimed to reconstruct the cathedral of Seville after the conquest of 1248 and until the construction of the new late Gothic temple in the 15th century. These publications have been fundamental for our research because they clarify the cathedral's spatial distribution in the 14th century, which is the first step in trying to locate or clarify any data on Pedro I's chapel. In addition to Professor Laguna

Paúl, we should also mention the work of Jiménez Martín (2007a, 2007c), Jiménez Martín and Pérez Peñaranda (1997), Almagro Gorbea (2007), Martínez de Aguirre (1995), Nogales Rincón (2009), and Pérez-Embid Wamba (2015).

In 2005, González Zymla shed some more light on the chapel. He wrote: "This chapel-pantheon seems to have been ordered to be built in the old cathedral of Seville, which was demolished years later to build the present one. Very little is known about this funerary chapel, except that it was besides a chapel dedicated to the Virgin of Granada, and that, as a work sponsored by the king, it was ambitious and well-endowed. The location of the chapel is unknown, although it can be assumed that it must have been around the main chapel, probably forming a sort of ambulatory"[2] (González Zymla 2005, p. 65). González Zymla's speculation about the location of the chapel is the first clear sign of interest in solving this question.

In his text, González Zymla located Pedro I's chapel next to the one dedicated to the Virgin of Granada. Thus, before continuing, this location must be clarified. As documented in the sketch of the Patio de los Naranjos made in 1803 and studied by Jiménez Martín, the chapel of the Virgin of Granada was located behind the NE exit of the Christianised Mosque, at the foot of the Giralda. The Chapel of the Virgin of Granada was originally dedicated to S. Christoval in the medieval temple (Jiménez Martín 2018, p. 268). González Zymla's proposal is interesting, but more arguments would be needed to consider this space the former chapel of Pedro I.

In 2006, Jiménez Martín (2007b, p. 37) wrote a major publication on the Christianised Mosque of Seville. In that publication, the author believed that Pedro I's Royal Chapel was built in the "Mudéjar" style, like the other buildings commissioned by the Castilian monarch. The last published study of this royal chapel also dates from 2006. Juan Carlos Ruiz Souza, considering the information in the testament and in the chronicle of López de Ayala, insisted that there was no doubt about the existence of the chapel, just as there is none for the Royal Chapel of Alfonso X, Fernando III, and Beatriz de Suabia. According to Ruiz Souza (2006, p. 12), the two chapels must have been very close to each other and towards the eastern wall of the former mosque.

After this brief review of the literature, it can be concluded that, to date, very little is known about Pedro I's Royal Chapel in Seville because of the lack of material and textual sources. However, two essential points must be taken from this historiography: first, according to all researchers, the chapel undoubtedly existed; second, it was most probably a work of "Mudéjar" architecture. Its location within the medieval church, however, remains an unresolved question.

## 2. The Christianised Mosque, the Space on Which the Pedro I's Royal Chapel Was Founded

To put forward any hypothesis about the location and morphology of Pedro I's Royal Chapel, it is essential to understand the formal organisation of Seville Cathedral during the 14th century.

Structurally, the 14th-century Seville Cathedral had the same layout as the Almohad mosque, which was completed in 1198. Caliph Abū Ya'qūb Yūsuf commissioned the project to the builder (*alarife*) Aḥmad b. Bāso. The poor state of the old mosque of Ibn A'dabbās and the limited space available to house the growing population of Seville led the caliph to construct the new building (González Cavero 2013, p. 215). A text by the Andalusian historian Ibn Ṣāḥib al-Salā translated by Fatima Roldan narrates the founding episode:

"During the month of Ramadan of this year [567H/1172C.], the Amir al-Mu'minin b. Amir al-Mu'minin [Abū Ya'qūb Yūsuf] began the layout of the site of this beautiful and excellent mosque. The houses inside the citadel were demolished, and the chief architect Ahmad b. Basu, his fellow architects from the city of Seville, all the architects of al-Andalus and those from the [central government] headquarters in Marrakesh, as well as those who practised in Fez and others [who came from the other] side [of the Strait], took charge of it. In this way, many

construction professionals were brought together in Seville, as well as specialised carpenters, sawyers and workers [all of them technicians related to] construction, each one expert and skilled in their speciality"[3]. (Roldán Castro 2002, p. 15)

Work began soon after the land expropriation, but it was halted for four years until 1182, when work resumed. In that year, the mosque started being used, even though the building was not concluded and works on the minaret had not even started. In 1184, Abū Yaʻqūb Yūsuf ordered the construction of the minaret but it was not finished until 1198 during the government of his son Abū Yūsuf Yaʻqūb al-Manṣūr and thanks to the designs of the architect Aʻlī al-Gumārī (Jiménez Martín 2007c, p. 133).

The Almohad mosque had a great formal unity because, unlike other mosques, such as the one in Córdoba, it was the result of a single project (Figure 1). Its structure had few concessions to decoration, following Almohad's aesthetic principles (Almagro Gorbea 2007, p. 98). It had a 113 × 135 m rectangular plan. The prayer hall was divided into seventeen naves, oriented north-south and separated by twelve bay arcades supported by large brick pillars. The naves were interrupted at the qibla wall by an east-west transversal nave (Almagro Gorbea 2007, p. 98).

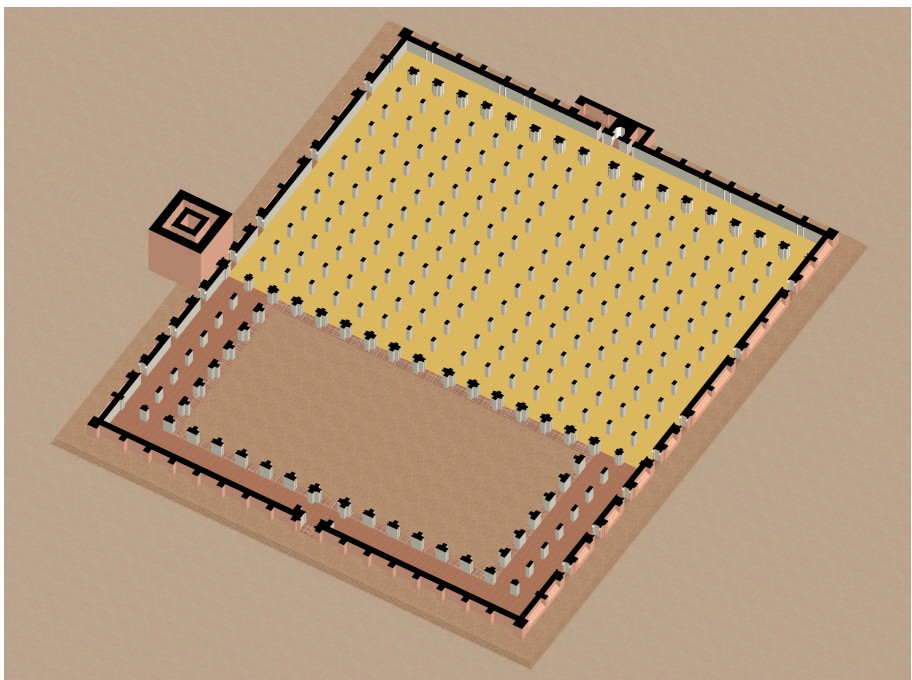

**Figure 1.** Plan of the Almohad mosque of Seville according to Almagro Gorbea (2007).

When Seville was conquered in 1248, the mosque became a Christian church under the title of Santa Maria following the consecration of Archbishop Gutierre (Dotor y Municio 1931, p. 5). From the 13th century onwards, the building was divided into two differentiated spaces: one with the main altar and dedicated to the clergy and the other reserved for the Royal Chapel of Alfonso X, at the service of the monarchy. Pablo Espinosa de los Monteros wrote:

> "The honourable, virtuous and wise King Don Alfonso, son of King Don Fernando, divided the church into two equal parts. On the west side, he placed the Blessed Sacrament and the Holy Image of Our Lady of the See (…) On the east side towards the tower, he made a Royal Chapel, leaving a clear passage around it, enclosing it with iron grilles, so that the view was unobstructed from all sides"[4]. (Espinosa de los Monteros 1635, p. 13)

Researchers such as Jiménez Martín (1984, p. 89) (Figure 2) or Almagro Gorbea (2007, p. 102) suggested possible reconstructions (Figures 2 and 3, respectively) of the spatial distribution of the Christianised Mosque; based on the words of Pablo Espinosa, the *Libro Blanco*[5], and the archaeological excavations, they have produced reconstructions of what the spatial distribution of the Christianised Mosque may have been. According to these authors, the entire eastern part of the structure was reserved for the Royal Chapel of Alfonso X, which, framed by a perimeter of grilles, may have been bordered by an ambulatory. The choir was located in the western half, with the area destined for the Cabildo (Jiménez Martín and Pérez Peñaranda 1997, p. 24). On both the southern and northern walls, i.e., occupying both the nave next to the *qibla* wall of the old mosque and the northern nave closest to the Patio de Naranjos, respectively, there was a row of chapels enclosed by grilles, in the same way as in the Cathedral of Córdoba. The perimeter of the Patio de Naranjos was also framed by a multitude of spaces reserved for private chapels[6].

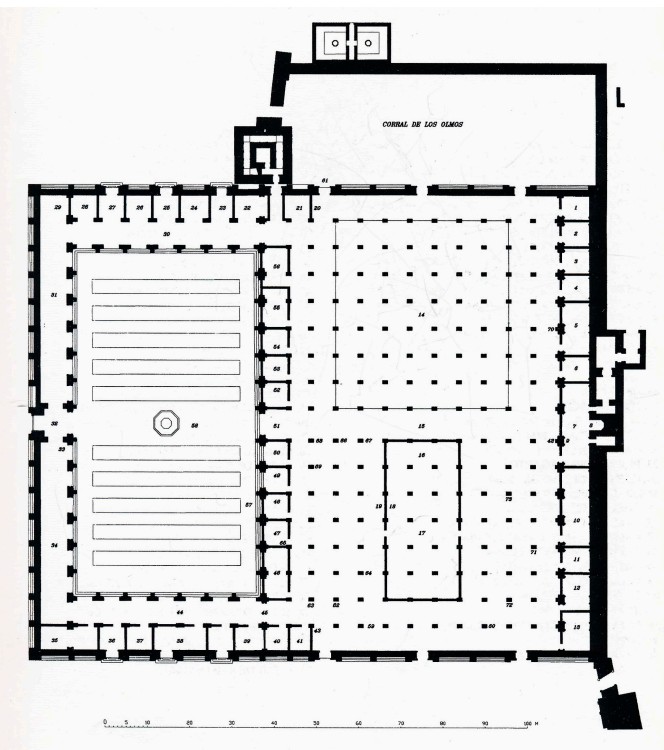

**Figure 2.** Plan of the Christianised Mosque of Seville according to Jiménez Martín and Pérez Peñaranda 1997.

According to Jiménez Martín, depending on their location within the church, these chapels belonged to nobles or ecclesiastics. The archbishops were placed in the western chapels of the cathedral, flanking the choir and forming an "escort of honour" (Jiménez Martín and Pérez Peñaranda 1997, p. 25). The aristocracy of the city of Seville and the royal officials acquired the eastern area, occupying those chapels that flanked the grandiose Royal Chapel of Alfonso X. Finally, the wealthier members of the bourgeoisie settled for the chapels in the Patio de Naranjos.

We can conclude that, within the space of the Christianised Mosque, monopolised by the Royal Chapel of Alfonso X and flanked by a very large number of clerical, aristocratic, and bourgeois chapels, there was not much space left for King Pedro I to build a great mausoleum worthy of his egomania and political ideals.

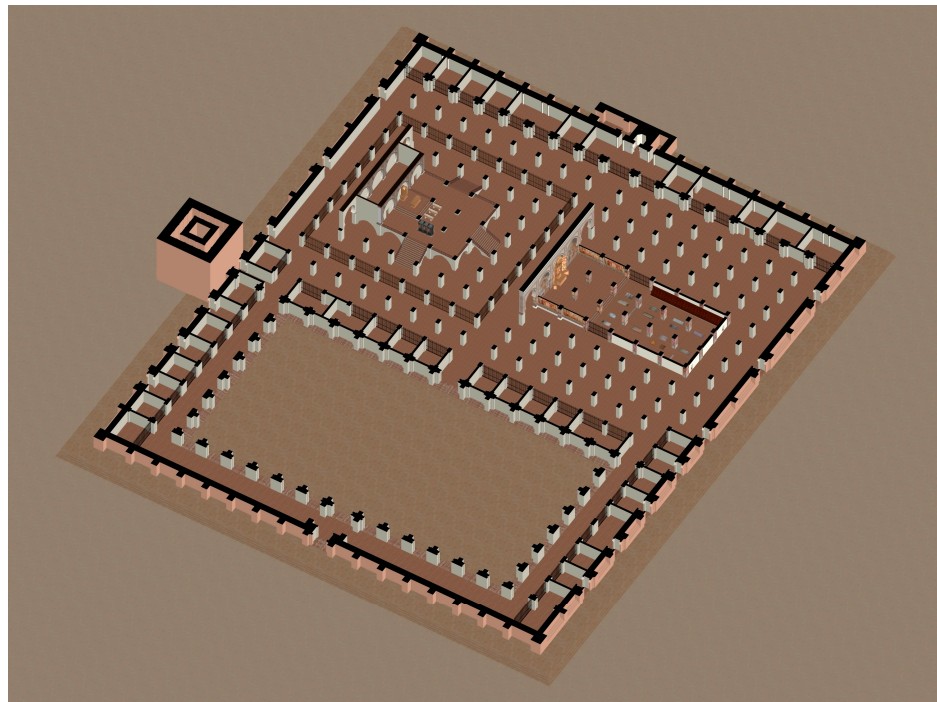

**Figure 3.** Plan of the Christianised Mosque of Seville according to Almagro Gorbea (2007).

### 3. Documentary Sources for the Study of Pedro I's Royal Chapel

Now that the scenario on which the Royal Chapel of Pedro I was planned has been understood, let us analyse the surviving data about this project (Figure 4). The first mention of its foundation is in the king's will, dated 18 November 1362:

> "And when I die, I order my body to be brought to Seville and buried in the new chapel that I now order to build [*fazer*]; and that they put Queen Doña Maria, my wife, "*de un cabo*" [measure] on my right, and "*del otro cabo*" on my left the Infant Don Alfonso, my first son and heir; and [I order] that they dress my corpse in the habit of Saint Francis and bury it in it [the chapel]"[7]. (López de Ayala 1877, p. 593)

Later in the same testament, he states:

> "I also order that my chapel, and that of my ancestor kings, and any other ornament in the church to be given to the chapel that I now order to build [*fazer*] here in Seville, where I am to be buried, and my said wife, and the said Infant my son, everything must be for said chapel. And [I order] that they give two paintings [*tablas*] and others that came from the chapel of the Kings, that are big, and others that are smaller in which the Signum Domini is [depicted]. And I order that they give three carpets, the best [that] I have, to be placed on the floor of said chapel where I am to be buried [...] And I also order to put twelve chaplains to sing masses continuously for my soul and the souls of the said Queen Doña María my wife and the Infant D. Alfonso, my son, in the said Church of Santa María, in the chapel that I order to build [*fazer*] where my body and those of the said Queen and Infante are to be buried: and that they cause them and that they fulfil all the masses and anniversaries to be said by the clergy and the orders and other things"[8]. (López de Ayala 1877, p. 593)

| SUMMARY OF THE DOCUMENTARY HISTORY OF THE CHAPEL OF PEDRO I IN THE CHRISTIANISED MOSQUE OF SEVILLE | |
|---|---|
| 1248 | Conquest of Seville by Fernando III of Castile |
| 1262 | Ruy López de Mendoza is buried in the Chapel of San Pedro |
| 1338 | Ruy Gonçález de Mançanedo acquire several spaces located to the south of the temple, including the Chapel of San Pedro |
| 24 August 1356 | An earthquake toppled the golden spheres of the *yamur* that had been placed on top of the minaret of the old Almohad mosque in Seville on 19 March 1198 |
| 24 August and 27 November 1357 | Pedro I plundered the crown jewels of Alfonso X and Beatrix of Swabia to finance the war against Aragon |
| 18 November 1362 | By testamentary order, Pedro I orders the foundation of a Royal Chapel |
| 18 November 1362 | By testamentary order, Pedro I granted 3000 maravedis for the repair of the tower of the cathedral of Santa María in Seville |
| 1363 | The body of Queen María de Padilla is transferred from Astudillo to Seville and provisionally buried in the Royal Chapel of Alfonso X |
| 1411 | Elaboration of the plan called "La Cuadra" and the *Libro Blanco* |
| 10 February 1433 | Juan II permits the demolition of the Royal Chapel of Alfonso X and the transfer of the royal bodies to the Patio de los Naranjos is organised |
| 13 June 1579 | Transfer of the royal bodies to the new Royal Chapel in Seville (including that of Queen María de Padilla) |

**Figure 4.** Summary of the documentary sources for the study of the Chapel of Pedro I.

This first document is irrefutable proof of the existence of at least an architectural project to build the chapel. Thanks to the will, it can be assured that in 1362, Pedro I had the firm intention of building his mausoleum, and of being buried together with his wife and son in a perfectly studied layout. It is also certain that the chapel was to have twelve chaplains to pray over their souls. Finally, the testament also documents the donation of numerous artworks to the chapel, such as several paintings from the Royal Chapel of Alfonso X, and three carpets to cover its floors.

The second contemporary reference to this chapel was written a year later by Pero López de Ayala. The chronicler explains how the body of María de Padilla was transferred from the convent of Santa Clara de Astudillo, where it had been buried two years earlier, to Seville. Since the Royal Chapel of Pedro I was not ready, she was buried provisionally in the Royal Chapel of Alfonso X:

> "And the King ordered Prelates, Noblemen and Ladies to go to Astudillo, where Doña María de Padilla was buried, and they brought her body very honourably to Seville, like a queen, and they buried her on the chapel of the kings, that is in the Church of Santa María of said city, until the king had another chapel built, near that of the kings, very beautiful, where the body was later buried"[9]. (López de Ayala 1877, p. 350)

This information is repeated in a very similar way by other later chroniclers, such as Jerónimo Zurita (1683)[10] and Vera y Figueroa (1647, p. 63)[11], but these authors must have taken the information from Ayala's chronicles, so their documentary value as direct sources

is irrelevant. We can therefore consider that the last direct reference we have to the chapel dates from 1363.

Ayala's chronicle gives us two fundamental pieces of information: first, that the chapel had not been completed by 1363, since the body of the queen was provisionally deposited in the Royal Chapel of Alfonso X. Second, that the body of María de Padilla was indeed later buried in the chapel founded by Pedro I: "*dó fué el dicho cuerpo después enterrado*". Ayala, who must have written the chronicles some years later, goes so far as to say that the chapel was "*muy fermosa*", which undoubtedly suggests that the chapel must have attained a certain architectural importance.

There is no documentary evidence of the destiny of Maria de Padilla's body after the chronicler attested to its inhumation in the Royal Chapel of Alfonso X in 1363 until 1579. In February 1433, when the works on the late Gothic cathedral were fully operational, King Juan II allowed the demolition of Alfonso X's chapel (Gestoso y Pérez 1984, p. 300). Consequently, the royal bodies were transferred to a provisional space in the Patio de Naranjos. In the documentation of the translation, there is no mention of María de Padilla's body, which would confirm López de Ayala's information: the queen's body was buried in the chapel of Pedro I at some point between 1363 and 1433. Even more, bearing in mind that Ayala saw the body of Maria de Padilla being moved to the chapel of Pedro I, it must have happened before he died in 1407.

Alonso Morgado wrote that on 13 June 1579, when Covarrubias finished the new Royal Chapel of Seville, the kings' bodies were transferred in solemn procession from their provisional burial in the Patio de Naranjos to the new Renaissance chapel[12]. The name of María de Padilla appears in Morgado's text, which indicates that, at some point in the 15th century, during the construction of the new late Gothic cathedral, her body was also transferred to the provisional burial site of the Patio de Naranjos, following the demolition of Pedro I's Royal Chapel.

The difficulties in locating the missing chapel appear when consulting the *Libro Blanco*. In 1411, an architectural plan known as "la Cuadra" and the *Libro Blanco*, a detailed description of the Christianised Mosque, were drawn up (Belmonte Fernández 2019). Both the plan and the description showed the formal layout of the building before its demolition for the construction of the new late Gothic cathedral (Jiménez Martín 2007b, p. 39). The plan is now lost because, according to tradition, Philip II moved it to the Alcázar of Madrid where it disappeared in the 1734 fire (Laguna Paúl 1998, p. 43). The *Libro Blanco* had better luck. It is not a book, but a series of parchments written by the prior Diego Martinez, which are bound in three volumes and stored in the Archive of the Cathedral of Seville (Jiménez Martín and Pérez Peñaranda 1997, p. 127). What is striking is that there is not a single reference to the Royal Chapel of Pedro I in the *Libro Blanco*, a fundamental source for the study of the Christianised Mosque. This problem is the wall against which all research on this building has crashed against. There must be an error either in the chronicle of López de Ayala or in the *Libro Blanco*. It is possible that the *Libro Blanco* likely referred to the chapel of Pedro I by its dedication, without mentioning the burial of Queen María de Padilla or the commission of the Castilian monarch.

There is one last source that could shed some light on the study of Pedro I's Royal Chapel. It is a statement written in 1635 by the priest Pablo Espinosa de los Monteros, which has so far gone unnoticed by all researchers. In a description of all the chapels in the already demolished Christianised Mosque, he writes about the former Chapel of San Pedro. Espinosa de los Monteros wrote:

> "In this Chapel there were two Altars, one was of Santa Maria de la Antigua, and another of Santa Maria de Alcobilla (this image is that of the Angustias) and the Altar of San Christobal. Dean Don Pedro Manuel and his mother Dona Berenguela Ponce were buried there. They founded a Chaplaincy and an Anniversary (...) In this Chapel the King Don Pedro was veiled with Doña Maria de Padilla, as it seems by a document [*instrumento*] of those times"[13]. (Espinosa de los Monteros 1635, p. 16)

Let us highlight the following point again: "*En esta Capillia se veló el Rey Don Pedro con Doña Maria de Padilla, según parece por vn instrumento de aquellos tiempos*". The text was again reported in the 19th century by José María Montoro (Montoro 1847, p. 185). Current historiography considers the information transmitted by Espinosa de los Monteros apocryphal. Julio Valdeón pointed out that "the most rigorous historical research has shown that there was no such matrimony" (Valdeón Baruque 2018). Juan B. Sitges also indicated that "the assertion that Don Pedro was veiled in the chapel of San Pedro in Seville Cathedral is devoid of all plausibility" (Sitges y Grifoll 1910, p. 391). In this sense, it is necessary to approach the source with a critical perspective. Nevertheless, in the absence of further data, it could be at least suggested that in Espinosa de los Monteros' eyes, the Chapel of San Pedro was a space in the church related to the medieval monarch. Perhaps there was some kind of oral tradition in the 17th century that led the writer to express this information.

### 4. Wrong Hypothesis: Pedro I's Royal Chapel on the Ancient *Maqṣūra*

Initially, we considered the idea that Pedro I's Royal Chapel was built in the space occupied by the chapel of San Pedro. However, as we are about to explain, this has led us to a dead end. The Chapel of San Pedro in the Christianised Mosque was installed on the old Almohad *maqṣūra*, i.e., in the centre of the southern wall (Figure 3). This was the most relevant space of the former mosque. Moreover, bearing in mind that the Royal Chapel of Alfonso X eclipsed the entire eastern space of the Christianised Mosque, the former *maqṣūra* was one of the most important spaces within the structure that was still free for the king to use.

The first documentary evidence of this Chapel of San Pedro in the *maqṣūra* dates from 1262, when Ruy López de Mendoza, the first Almirante Mayor de la Mar, was buried there (Jiménez Martín 2007b). In 1338, a new documentary source states that Rui González Manzanedo, commander of Montemolín, acquired a set of chapels in the southern part of the church, including the Chapel of San Pedro, for 8600 *maravedíes*. Thus, by 1362, the Chapel of San Pedro was in the *maqṣūra*, had the burial of Ruy López de Mendoza, and was in the patrimonial possession of Rui González Manzanedo or his heirs. Pedro I, having been able to plunder the jewels of his royal ancestors to finance the war against Aragon, would have had no qualms about expropriating the possessions of a nobleman, nor about moving his tombs.

Nevertheless, as the *Libro Blanco* pointed out, at the beginning of the 15th century, several noblemen were still buried in the Chapel of San Pedro. It is the case of Don Alfonso Sánchez de Çea, Leonor de Vargas, or the very same Rui González Manzanedo, who has remained in the chapel since the 14th century (Belmonte Fernández 2016). The hypothesis, although very suggestive, must be rejected. If it ever occurred to Pedro I to use the former *maqṣūra*, it did not materialise this way.

### 5. Conclusions: A Dead End

The almost total absence of historical data on the Royal Chapel built by Pedro I in Seville's Christianised Mosque makes it almost impossible to conduct a study that would clarify its constructive history and location. What is clear is that this building did exist. López de Ayala testified how the body of María de Padilla was buried in the chapel, and how beautiful it was ("*muy fermosa*"). Moreover, when, in 1433, the tombs were moved from the Royal Chapel of Alfonso X to a temporary space in the Patio de los Naranjos, the body of María de Padilla was no longer there. This again supports the idea that the chapel existed.

We have tried to propose a hypothesis about its location in the Chapel of San Pedro, the former Almohad *maqṣūra*, but 15th-century documentation has disproved it. We are at a dead end. Nevertheless, we consider this article fundamental to put on the table all the existing documentation on this mausoleum, to prevent it from falling into oblivion. We hope that in the future new documentation will appear that can shed light on the location

of this burial place where Pedro I planned to rest together with María de Padilla and their son Alfonso.

**Funding:** This research received no external funding.

**Data Availability Statement:** No new data were created or analyzed in this study. Data sharing is not applicable to this article.

**Acknowledgments:** I would like to thank Concepción Abad Castro, without whose guidance this article would not have been possible.

**Conflicts of Interest:** The author declares no conflict of interest.

## Notes

1    Original in Spanish: "Fundó en Sevilla una insigne capilla, a quien enriqueció de ornamentos, (hizo liberales donaciones, dotó de doce capellanías que siempre sufragaban su alma)".

2    Original in Spanish: "Tal capilla-panteón parece ser que fue mandada construir en la catedral vieja de Sevilla, demolida años más tarde para edificar la actual. Se sabe muy poco acerca de esta desaparecida capilla funeraria, apenas que estaba yuxtapuesta a una capilla dedicada a la Virgen de Granada, y que, como obra patrocinada por el rey, era ambiciosa y bien dotada de rentas. La ubicación de la capilla es ignota, aunque cabe suponer que debió estar en torno a la capilla mayor, formando probablemente una suerte de girola".

3    In Spanish: "Durante el mes de ramadán de este año [567H/1172C.], el emir al-mu´minin b. amir al-mu´minin (Abū Ya'qūb Yūsuf) comenzó el trazado del emplazamiento de esta hermosa y excelsa mezquita aljama. Se derribaron viviendas que había en el interior de la alcazaba y se encargaron de ello el arquitecto mayor Ahmad b. Basu, sus colegas arquitectos de la ciudad de Sevilla, todos los arquitectos de al-Andalus y los de la sede [central del gobierno], Marraquech, así como los que ejercían en Fez y otros [que procedían de la otra] orilla [del Estrecho]. De esta forma se reunió en Sevilla un número importante de profesionales de la construcción, así como carpinteros especializados, aserradores y operarios [todos ellos técnicos relacionados con la] construcción, expertos y hábiles cada uno en su especialidad".

4    In Spanish: "el honrado y virtuoso y sabio Rey Don Alfonso hijo del Rey Don Fernando partió la iglesia en dos partes iguales. En la parte del Poniente se puso el Santissimo Sacramento, y la Santa Imagen de Nuestra Señora de la Sede (…) La parte del Oriente hazia la Torre, hizo Capilla Real, dexando franco passo al rededor della, para que se penetrase la vista por todas partes, cercándola de rejas de hierro".

5    The Libro Blanco is a compilation of data on the old Christianised mosque dated 1411. We will speak about this source later in this article.

6    More on funerary chapels: (Pérez-Embid Wamba 2015).

7    In Spanish: "E quando finamiento de mi acaescier, mando que el mi cuerpo que sea traido á Sevilla, é que sea enterrado en la capiella nueva que yo agora mando facer; é que pongan la Reyna Doña Maria mi muger del un cabo á la mano derecha, é del otro cabo a la mano esquierda al Infant Don Alfonso mi fijo primero heredero; é que vistan el mi cuerpo de abito de Sant Franco, e lo entierren en él".

8    In Spanish: "Otro si mando que la mia capiella, e la que fue de los Reyes onde yo vengo, e cualesquier otros ornamentos de la Eglisia que yo tengo lo den todo a la capiella que yo agora mando facer aquí en Sevilla, do he de estar enterrado yo, e la dicha Reina mi muge, e el dicho Infant mi fijo, que sea todo para la dicha capiella: e quel den dos pares de tablas que están y unas que fueron de la capiella de los Reyes, que son grandes, e otras que son mas pequeñas en que está Signum Domini. E mando que den tres alombras de las mejores que tengo que ponga por suelo en la dicha capiella do he de estar enterrado [...] E otro si mando que pongan doce capellanes que canten continuadamente misas por mi alma e por las almas de la dicha Reina doña María mi mujer e del Infant D. Alfonso mi fijo en la dicha Eglisia de santa María, en la capiella que yo fago facer do han de estar enterrados el mi cuerpo e los de la dicha Reina e Infante: e que las causen e que lo cumplan todo asi misas como aniversarios que han a decir los clérigos e las ordenes e las otras cosas".

9    In Spanish: "E luego ordenó el Rey Perlados, é Caballeros, é Dueñas que fuesen á Estudillo, do yacia Doña Maria de Padilla enterrada, é traxieron su cuerpo muy honradamente a Sevilla, asi como de Reyna, é soterráronle en la capilla de los Reyes, que es en la Iglesia de Sancta Maria de la dicha cibdad, fasta que el Rey fizo facer otra capilla cerca de aquella capilla de los Reyes, muy fermosa, dó fué el dicho cuerpo después enterrado".

10    "e soterraronla en la Capilla de los Reyes, fasta que el rey fizo otra capilla cerca de aquella de los Reyes do fue el dicho cuerpo despues puesto".

11    "Y en esta conformidad se dispuso proporcionando acompañamiento que fue a Asudillo y trajo el cuerpo de Doña María de Padilla a la Capilla de los Reyes de Sevilla: Allí estuvo hasta que se labró otra donde fue trasladado y desde este día corrientemente llamada reina de Castilla y Leon y príncipe y infantes sus hijos".

12 (Morgado 1587, p. 107) In Spanish: "Se juntaron en aquella Capilla, donde estavan los Cuerpos Reales, el arzobispo desta ciudad Don Christobal de Rojas, de sancta memoria, el Regente de la Audiencia Real de Sevilla y su asistente, en día sábado treze dias de junio del año de mil y quinientos y setenta y nueve a las siete de la Tarde [...] Los quales todos descubrieron alli el cuerpo del glorioso San Leadro, y dos imagines muy antiguas, y muy devotas, de nuestra Señora, el Cuerpo del Santo Rey Don Fernando, y de la serenisima reyna Doña Beatriz su muger, y del Rey Don Alfonso el Sabio su hijo, el de Doña Maria de Padilla, y Cuerpos de los Infantes Don Alfonso, Don Pedro y Don Fadrique, Maestre de Santiago...".

13 In Spanish: "En esta Capilla vuo dos Altares, vno era de Santa Maria de la Antigua, y otro de Santa Maria de Alcobilla (esta imagen es la de las Angustias) y el Altar de san Christoual. Estuuieron enterrados en ella el Dean Don Pedro Manuel; y su madre Dona Berenguela Ponce. Fundaron una Capellanía y vn Aniversario: Y el canonigo Alonso López, que fundó otra Capellanía y otro Aniuersario, y la procession de san Elifonso. En esta Capillia se veló el Rey Don Pedro con Doña Maria de Padilla, según parece por vn instrumento de aquellos tiempos".

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
