# Peer review of "The Royal Chapel of Pedro I of Castile in the Christianised Mosque of Seville"

_arts, 2018_

Round 1
Reviewer 1 Report
Comments and Suggestions for Authors
The article offers a state of the art, with a reference to the main testimonies about the royal chapel of King Pedro I of Castile. The cited bibliography is adequate. The hypothesis, although it may be suggestive, is still a merely hypothetical proposal, which should be reinforced, at least, in relation to two aspects:
- Current historiography considers as apocryphal news the information transmitted by Espinosa de los Monteros, which places the vigils of King Pedro I and María de Molina in the chapel of San Pedro, as Julio Valdeón points out, indicating that “la más rigurosa investigación histórica ha puesto de manifiesto que no existió ese matrimonio” (Valdeón Baruque, Julio: “María de Padilla”, Diccionario Biográfico electrónico (DB~e), Madrid: Real Academia de la Historia, 2018 [en línea], disponible en http://dbe.rah.es/biografias/7727/maria-de-padilla) or Juan B. Sitges, who indicates that “la afirmación de que Don Pedro se veló en la capilla de San Pedro de la Catedral de Sevilla está destituida de toda verosimilitud”, Sitges y Grifoll, Juan Bautista: Las mujeres del rey Don Pedro I de Castilla, Madrid: Sucesores de Rivadeneyra, 1910, p. 391. In this sense, it would be necessary to offer a critical perspective of these informations.
- The privatization of sacred space by royalty is a fact throughout the late medieval period (for example, the cases of the cathedrals of Seville or Toledo), but there are some problematic informations (which proceded from the Libro Blanco) that inform us about the existence of several burials in the chapel of San Pedro at the beginning of the 15th century: “Don Alfonso Sánchez de Çea, maestrescuela de la iglesia, tiene vna sepoltura en esta capilla de sant Pedro” (nº 93); “En esta capilla del Antigua, que se llama de sant Pedro, está enterrada Leonor de Vargas, muger que fue de Diego de Yllescas e fija de Alfonso Garçía de Vargas, la qual dexó por su heredero a Antón Garçía de Santlúcar” (nº 95);“Ítem ha de auer la Obra de la iglesia después de los días de Alfonso López, canónigo, dozientos maravedís de cada anno por el dote de la capilla de sant Pedro, do él está enterrado”, nº. 346. In fact, Rui González Manzanedo remains in the Chapel of San Pedro at the beginning of the 15th century: "Ruy Gonçález Mançanedo en el mes de deziembre, X maravedís. Ítem dan a otra memoria por donna Berenguella, muger del dicho don Ruy Gonçález, en el mes de deziembre, X maravedís. E todos estos están enterrados en esta capilla [de San Pedro] (nº 83). (Belmonte Fernández, Diego: Organizar, administrar, recordar. El libro blanco y el libro de dotaciones de la Catedral de Sevilla, doctoral thesisl, Universidad de Sevilla. Departamento de Historia Medieval y Ciencias y Técnicas Historiográficas, 2016. URL: https://idus.us.es/handle/11441/44471). This information should be taken into account when offering a reasoned argument for the proposed hypothesis.
In our opinion, three aspects must be corrected:
- Espinosa de los Monteros refers to the fact that “en esta Capilla se veló el Rey Don Pedro con Doña Maria de Padilla”. The author says that the chapel “was the place where the funeral and vigil of Queen María de Padilla took place” (line 251). It is incorrect, since “velarse con” is an unequivocal reference to a marriage ceremony, not a funerary one.
- The Figure 5, where the tombs are represented in a perpendicular position to the main altar of the Virgen de la Alcobilla, does not correspond with what the author indicates in the text, when pointing out that “Peter I would occupy the central space, with his feet facing the altar as in most mausoleums” (lins. 206-297). In the Figure 5, the king's feet are not facing the altar of the Virgen de la Alcobilla. In this sente, either the text or the figure must be corrected.
- "Ruy Gonçález de Mançanedo" (lins. 274, 277). The name of this person must be modernized, as it is usual in historiograhpy. In this sense, the correct form is "Rui González Manzanedo".
Author Response
Thank you very much for your response and your review. The truth is that your arguments have completely dismantled the article for me. Especially those concerning the burials in the chapel of San Pedro throughout the 15th century. I have admitted defeat and have accepted the hypothesis of the Chapel of San Pedro as erroneous. Nevertheless, even if it is a dead end, I still consider this article fundamental to put on the table all the information available on this building, which undoubtedly had to exist, and thus prevent it from being forgotten. Also, as recommended, I have corrected the name "Rui González Manzanedo". Again, thank you very much for your suggestions, which I greatly appreciate.
Reviewer 2 Report
Comments and Suggestions for Authors
See attached document

See attached document
Author Response
I appreciate your comments and corrections which I implemented consciously. As you will see in the paper, I had to make great changes due to the corrections of the first reviewer. He has given me solid arguments that disprove the idea that Pedro I´s chapel was displayed in the former maqsura. But I still consider the publication important to gather all the information on the building and prevent it from falling into oblivion. Therein lies the value of my text.
I also want to thank you for the language corrections. It is great to count on a native reviewer. Especially, as I am not currently involved in any project that can finance a professional language review, I really appreciate it. Please do not hesitate to let me know if you find any other errors.
Reviewer 3 Report
Comments and Suggestions for Authors
Hello.
Thank you for letting me read your essay. I hope you will find my feedback helpful. Please understand I am reading this critically and I'm not shying away from what I believe you would like to hear; Namely, an honest critique and helpful source of feedback. I will provide it chronologically from beginning to end, follow in your essay paragraphs.
I'm going to start by pointing out that humanities papers should not have numbers. Numbers are found in scientific articles and statements, and it looks very amateur to suggest numbers are needed for a reader to understand an argument that is about to be laid out.
Your introduction paragraph needs to have a stronger start, as the first three sentences are basically the same and you only need one of them. Also, when you make this statement you immediately should have a footnote pointing out the few times it has been referenced. Also, you have it organized in a strange way. If you're making the statement that historians have not shed much light on this or discussed it, you need to explain why you're separating the paragraphs the way you do. Your first paragraph ends with you talking about it being mentioned again in the 20th century, yet the next paragraph starts with the 20th century, with no explanation of how this information is any different than the earlier century references. Why don't you just put it all together in one single paragraph and make sure each thing is being said regarding a different point or combine the references to the same points. You could also divide it by centuries but make it clear that your first paragraph is from the 17th to the 19th centuries. Then start your next paragraph by introducing the 20th century documents. Then bring the next paragraph into our age now. Honestly, the beginning of this paper should be substantially clearer than it is and I shouldn't have this many comments this early on about simply presenting information, not repeating information, and having footnotes be presented immediately when you make a statement about many different people discussing this topic rarely discussed.
As a note, your abstract aside, your reader has no idea what this paper is about because you have no introduction paragraph that states what you intend to do regarding research and an argument. All I know is that you continually state nobody has discussed the work in full, and this is still going on into the second page. It is not enough to say something has barely been researched. That's basically the premise for most graduate papers. You need to state, in the imperative and affirmatively, what you are contributing from the onset. A paper is not supposed to be an unfolding mystery.
I am now only, on page 3, understanding that your hypothesis is about the location and morphology of the chapel. Again, it is not the responsibility of the abstract to state that, but your responsibility in the very beginning of the essay to clarify its purpose and goals. That said, your historical documentation is good.
You should really avoid language that is personalized, such as the word “we”, that you conclude your second section in on page 6. It's completely antiquated. Just speak with authority directly and succinctly.
Jumping ahead to page 9, you really need to clarify what the problems are that you're referring to. Your research is impressive and quite exhaustive, which speaks to your skills in that regard. Still, you have to remind your reader, throughout the essay, why you are providing these citations and quotes. You're leaving too much open to the reader and this feels like an annotated bibliography more than it does an actual research paper right now.
When you begin section 4 you really should avoid using the word “interesting”. When someone says something is interesting, the statement alone is never that [interesting]. Instead, try explaining specifically what you mean, what you find interesting. It is a blanket term used by undergraduate students, almost always unsuccessfully.
Regarding your conclusion, you really shouldn't begin the paragraph by saying “it can be concluded”. You have spent a lot of time and research coming to a conclusion. The idea is that, after reading the essay, your assertions have been proven. You absolutely should avoid this tedious language. It is an outdated way of writing, and you just need to start with the other part of the sentence that says “the almost total absence...”.
Overall, this is a very impressive paper regarding your interest in the topic and your research. You have a lot of editing you need to do regarding hesitant phrases, clauses that are unnecessary, and speculative discourse. Have confidence and just state things directly and clearly as if they are already facts, because that's what you are stating by even creating this; you are providing facts to prove your argument. You need to find out why you are interested in debating the location and change of this chapel, or at least make your reader more interested. The paper seems to solely be about the location and history and has very little art historical discussions about style, architectural movements, or cultural and artistic visual influences in its form. I think that would round it out and make it a more cohesive paper, rather than a listing of primary documents.
You have all the research done, in many ways, and now you just need to turn this into an essay that is saying something beyond that you can prove the topic hasn't been studied and questions still exist. Perhaps you should step back from this and think about why you care about this and get to the heart of the substance of your argument, aside from your impressive research and use of primary sources. How is it engaging and engrossing? Think about that.
Comments on the Quality of English LanguageIt is not so much about the quality of the language, but its syntax.
Author Response
Thank you very much for your response and your review. As you will see in the paper, I had to make great changes due to the corrections of the first reviewer. He has given me solid arguments that disprove the idea that Pedro I´s chapel was displayed in the former maqsura. But I still consider the publication important to gather all the information on the building and prevent it from falling into oblivion. Therein lies the value of my text.
Following your recommendations I have included an introduction before starting with the state-of-the-art. I believe it is quite repetitive if you read the abstract before, but I agree, it is not the abstract role to introduce the article. I have also insisted on the purpose of the article in parts such as p. 9.
Reviewer 4 Report
Comments and Suggestions for Authors
I appreciate the effort put into this article. Specific, well-chosen topic, as well as professionally completed. Good selection of archival sources, good bibliography, and - first and foremost - good, logical construction. The main subject is a kind of gloss on the history of Gothic architecture but it doesn't change the fact that the text is valuable and useful for scholars interested in similar topics.
Author Response
Thank you very much for your response and your review. I really appreciate your comments. As you will see in the paper, I had to make great changes due to the corrections of the first reviewer. He has given me solid arguments that disprove the idea that Pedro I´s chapel was displayed in the former maqsura. But I still consider the publication important to gather all the information on the building and prevent it from falling into oblivion. Therein lies the value of my text.
Round 2
Reviewer 1 Report
Comments and Suggestions for Authors
As simple recommendations and while the hypothesis of locating the royal chapel of Pedro I at the chapel of San Pedro has been discarted, we propose to erase at "Figure 4. Summary of the documentary sources for the study of the Chapel of Pedro" the references to de Chapel of San Pedro. Also, in the area of Humanities it is strange to present wrong hypotheses, so it is proposed to eliminate the section "Wrong hypothesis: Pedro I's Royal Chapel on the ancient Maqṣūra" and present a simple state of the art
Reviewer 3 Report
Comments and Suggestions for Authors
The author showed thought and effort in revising. The essay is much stronger. Job well done!